# Ratio between Height and Thickness of the Buccal Tissues: A Pilot Study on 32 Single Implants

**DOI:** 10.3390/dj7020040

**Published:** 2019-04-02

**Authors:** Davide Farronato, Mattia Manfredini, Francesco Mangano, Giada Goffredo, Marco Colombo, Pietro Pasini, Andrea Orsina, Marco Farronato

**Affiliations:** 1School of Medicine and Surgery, University of Insubria, Via G. Piatti 10, 21100 Varese, Italy; davide@farronato.it (D.F.); pm.pasini@gmail.com (P.P.); aa.orsina@gmail.com (A.O.); 2C.so Europa 10, 20122 Milan, Italy; mattiamanfredinidr@gmail.com (M.M.); giada_goffredo@hotmail.it (G.G.); 3Department of Prevention and Communal Dentistry, Sechenov First Moscow State Medical University, 119992 Moscow, Russia; 4V.le Luigi Borri 133, 2100 Varese, Italy; m.colombo.dds@gmail.com; 5Department of Orthodontics, School of Dentistry, University of Milan, Via Commenda 10, 20154 Milan, Italy; marcofarronato@msn.com

**Keywords:** implants, aesthetics, tissue height, tissue thickness, conical implant–abutment connection, platform switching, success

## Abstract

*Background*: Previous studies have suggested that mucosal height is related to the bone level and soft tissue thickness. The purpose of this pilot study was to investigate the ratio between the height and width of the tissues around single implants with a conical connection and platform switching. *Methods:* All patients receiving single implants (Anyridge^®^, MegaGen, Gyeongbuk, South Korea) and restored with single crowns, in a three-month period, were included in this study. After a provisionalization of 12 months, precision impressions were taken and stone casts were poured for measurements. For each implant, two values were collected at the buccal site: the mucosal height (MH), calculated from the vestibular shoulder of the implant to the upper gingival margin of the supra-implant tissue; and the mucosal thickness (MT), calculated from the vestibular shoulder of the analogue to the external mucosa point perpendicular to the implant major axis. Mean, standard deviation (SD), and confidence intervals (CI 95%) for MH and MT, as well as their ratios, were calculated. Correlation between MH and MT was assessed by Pearson’s correlation coefficient, with significance level set at 0.05. *Results*: 32 single Anyridge^®^ implants were eligible for this evaluation. The mean MH was 3.44 mm (±1.28), the mean MT was 3.29 (±1.46). The average of the ratio between MH and MT of the supra-implant mucosa was therefore 1:1.19 (±0.55). The relation between MH and MT was statistically significant at *p* ≤ 0.01 (Pearson two-tailed 95% CI). *Conclusions:* Our study found a constant relationship between width and height of the peri-implant mucosa. However, our results are different from those of Nozawa et al., who found a ratio of 1:1.5 between height and thickness of the peri-implant tissues. This may be determined by the different sample and follow-up period, as well as by the implants used in our study.

## 1. Introduction

Nowadays, one of the most challenging tasks in implant dentistry is to fulfill the aesthetic expectations of patients [1,2]. In the last few years, in fact, attention has shifted from the achievement of osseointegration (i.e., the formation of a direct interface between implant and bone without intervening soft tissue) and therefore from implant function, to the long-term aesthetic outcomes [3]. As a result, the success criteria in implantology have changed, according to the evolution of implant design and patients’ requests [4]. Today, the parameters for clinical success published by Alberksson et al. can no longer be considered the reference standard in implantology [5]. An osseointegrated implant under functional load, but which does not integrate perfectly in the patient’s oral cavity from the aesthetic point of view, can no longer be considered acceptable. This is particularly true for single-implant restorations, where aesthetic outcome plays a fundamental role and is a need for the patient [6].

Since the aesthetic result has become of primary importance in implant rehabilitation, numerous indices have been developed in order to evaluate the aesthetics of a single restoration along time [7,8,9,10]. In 2005, Fürhauser et al. proposed the pink esthetic score (PES) in order to evaluate the aesthetic outcome of implant-supported single restorations. This index was based on the evaluation of seven soft-tissue parameters [7]. In 2009, Belser et al. further developed the Furhauser concepts, proposing a new index called the pink esthetic/white esthetic score (PES/WES); this index takes into account not only the soft tissues around the implant-supported crown, but also prosthetic parameters [8]. Both of these evaluations are based on photographs and models and give credit to the peri-implant soft tissue frame, showing its importance to the final aesthetic outcome [7,8,9]. More recently, a new three-dimensional (3D) method has been proposed to evaluate the stability of peri-implant tissues along time [10]; this method is based on the superimposition of intraoral scans taken at different follow-up times, in a reverse engineering software, to evaluate any possible modification at a micro-metric level [10]. 

Nevertheless, what are the factors that determine the aesthetic result of an implant treatment? Undoubtedly, among the factors that determine the aesthetic quality of an implant-supported restoration, there are the 3D position of the implant (also intended as depth and inclination) [3,6,9,11] as well as the quantity and quality of the hard and soft tissues, and their variations over time [11,12,13,14]. Important changes occur on hard and soft tissue following dental extractions and continue until 2 years after implant insertion and prosthetic rehabilitation [11,12]. These modifications are related to vascular support and mechanical function [12,13]. In particular, soft tissue should be considered adapting to the condition of the local setting [12,13,14,15]. Buser et al. in 2004 analysed the aesthetic result in relation to 3D implant positioning, focusing their research on the ideal amount of hard tissue that should be maintained around a dental implant [16]. Wang et al. in 2011 focused on soft-tissue thickness around implant-supported restorations while discussing distinct differences between thin and thick tissue biotypes [17]. Soft tissue biotype is an important parameter to consider in the aesthetic implant restoration, improving immediate implant success and preventing future mucosal recession [17]. The management of soft tissues during surgical and prosthetic phases is therefore essential [18]. Furthermore, for Wang et al., the prosthetic emergency profile is important to manage soft tissue around a dental implant; particularly, the proper contour and maintenance of the right amount of tissue is considered crucial [2]. 

In any case, the aesthetic outcome of an implant-supported restoration seems to be strictly related to buccal hard- and soft-tissue height [19,20,21]. Clinical and histological studies on natural teeth have correlated height with thickness of soft tissues [20,21]. With dental implants, Bengazi et al. suggested that the height of the mucosa is related to the bone level and soft tissue thickness [22]. In 1996, Wennstrom et al. pointed out the relation between the height and the width of the free gingiva around natural teeth, measuring a ratio of 1.5:1, respectively [23]. 

Following Wennstrom’s findings, Nozawa et al. analysed the volume of soft tissue around internal hexagon implants with a flat-to-flat connection (Frialit-2^®^ Dentsply, York, PA, USA) in 2006 [24]. They investigated 14 patients with single implant-supported restorations after an average period of 3 years and 5 months. The ratio between the height and width of implant found by Nozawa et al. was, respectively, 1:1.5. It was supposed that the changes of the height were regulated by the width [24]; this ratio, moreover, represents the volumetric tendency to maturation of the peri-implant soft tissue. Especially the horizontal thickness at the connection level seems to be a strong influencing parameter, capable to prevent recession during tissue maturation and remodelling [24]. 

The effect of platform switching, found by Lazzara et al. in 2006 [25], is nowadays well known in the literature [25]. Implants incorporating platform switching provide a greater quantity of connective tissue over the shoulder, which prevents bone resorption after healing of screw positioning [25].

The purpose of this study was to verify the findings of Nozawa et al. in a larger implant sample, and to understand if they are valid one year after the provisional restoration. Hence, the present research aimed to find correlations between the height and thickness of the buccal periimplant tissue, when using tapered implants with a 5-mm-deep conical connection (10°) and integrated platform switching. In particular, the hypothesis to be verified was whether the ratio between the height and the thickness of the tissues was stable in all the analyzed models, as demonstrated by Nozawa. Furthermore, we wanted to verify whether the dimensional relationship between the tissues was the same as Nozawa, or was represented differently, since the implant system used here was different. 

## 2. Methods

### 2.1. Patient Selection

All patients treated with single implants (Anyridge^®^, MegaGen, Gyeongbuk, South Korea) and restored with single crowns, in a three-month period between May and July 2015, were considered for inclusion in the present study. These implants featured a knife-edge thread design [26] and a nanostructured, calcium-incorporated surface [27,28] with a 5-mm-deep conical connection (10°) combined with an internal hexagon and integrated platform switching. Further inclusion criteria were age between 18 and 80 years, good systemic health, good oral hygiene (achieved through professional oral hygiene sessions, twice per year, and daily domestic care), and consent to participate in this data collection study, attending all periodic examinations. Conversely, exclusion criteria were pregnancy, lactation, heavy smoking habit (more than 20 cigarettes/day), severe medical conditions that could affect periodontal health, use of drugs correlated to periodontal hypertrophy, as well as insufficient bone volume that required major/minor hard tissue augmentation procedures before implant placement. The study was conducted in accordance with the principles outlined in the Declaration of Helsinki on clinical research involving human subjects, 1975 (revised in 2008), and it has been approved by the ethics committee of Insubria University with number #826-0034086 ‘‘Studies on the survival and the surgical-prosthetic success of dental implants: Influence of the implant–abutment connection”. 

### 2.2. Surgical and Prosthetic Procedures

All implants were placed slightly subcrestally, as recommended by the implant manufacturer [28,29]. A provisional screw-retained prosthetic restoration was placed after 4 months from surgery and left for 12 months, waiting for tissue maturation. After 12 months, provisional restorations were removed and a silicone precision impression (Aquasil®, Dentsply Sirona, York, PN, USA) was taken with an individualised transfer technique. The emergence profile of the provisional restoration was reproduced in the final metal-ceramic restoration through the individualised transfer technique (Figure 1). 

Stone casts were poured for measurements. All measurements required were collected with a calibre on the dental stone cast of the final crown (resolution 10^−2^ mm), by the same calibrated experienced operator. Patients with damaged or missing stone casts, on which it was impossible to collect measurements, were excluded. Following Nozawa’s measurements [24], two values were collected for each implant in the buccal area. 

The height of the peri-implant mucosa (mucosal height, MH) was calculated from the vestibular shoulder of the implant to the upper gingival margin of the supra-implant tissue in buccal position. This measurement (MH) corresponds to the depth of the implant referred to the most coronal point of the buccal mucosa measured according to the main implant axis. This measure is acquired with the posterior part of the caliber dedicated to the depth measurement. The inner tip is slided inward until it reaches the analogue shoulder and the rest of the calibre is stranded in contact with the coronal gingival margin at the most buccal and apical level. 

The width of the peri-implant mucosa (mucosal thickness, MT) was calculated from the vestibular shoulder of the analogue to the external mucosa point perpendicular to the implant major axis. In order to acquire this value, one of the tips of the calibre was placed in contact with the analogue shoulder and it was closed until the other tip could get in contact with the facial mucosa (Figure 2).

Statistical analysis was performed with SPSS^®^ (SPSS Inc, Chicago, IL, USA). Mean and SD of peri-implant mucosal height and width (MH, MT), as well as their ratio, were calculated. Correlation between MH and MT was assessed by Pearson’s correlation coefficient with significance level set at 0.05 and a confidence interval (CI) set at 95%. 

## 3. Results

In total, 24 patients were considered for inclusion in this study. These patients had been treated with 36 single Anyridge^®^ implants, but only 32 of these were eligible for the evaluation; four implants, in fact, were excluded due to lack of precision or bubbles in the measurement area on dental stone cast. As a consequence, 20 patients (5 males and 15 females, mean age at implant placement 57 years (±, range 37–76 years) were included in this study. These patients were installed with 32 implants (14 maxilla, 18 mandible). The implant positions were as follows: one incisor, one cuspid, 11 premolars and 19 molars (Table 1). The average value of the MH was 3.44 mm (SD 1.28), while the MT mean value was 3.29 (SD 1.46). The obtained average of the ratio between MH and MT of the supra-implant mucosa was 1:1.19 with an SD of 0.55. The relation between MH and MT was statistically significant at *p* ≤ 0.01 (Pearson two-tailed 95% CI). 

In the present analysis, multiple diameters of implants and positions are considered and evaluated. The collection of these measures and positions depends from the need of the patients that casually presented in the office during the present perspective evaluation. The system in use presents a common prosthetic platform and the diameter occupied by the prosthetic structures in the area adjacent to the connection is common to any different implant diameter and rehabilitations. This could be one of the reasons why no correlation was found between the MH/MT ratio and: the implant diameter with *p* = 0.14 (Pearson two tailed, 95% C.I.) and again between the MH/MT ratio and the implant position with *p* = 0.2 (Pearson two tailed, 95% C.I.) as well as the individual MH and MT values are respectively not significantly correlated with diameter at *p* = 0.57 and *p* = 0.3 (Pearson two tailed, 95% C.I.) and position at *p* = 0.78 and 0.25 (Pearson two tailed, 95% C.I.)

## 4. Discussion

Today, aesthetics is one of the factors that distinguish the success of implant-prosthetic rehabilitation, especially in the case of a single implant. For this reason, the literature has focused in recent years on the research of the parameters determining aesthetic success; among these we must remember the correct 3D implant positioning [3,6,9,11,16] and the quantity and quality of peri-implant hard and soft tissues [11,12,13,14,17,19], and their stability in time. Last but not least, of fundamental importance is the emergence profile of the implant-prosthetic restoration [2], which is decisive. In this context, it is essential to have an adequate height of peri-implant soft tissues, which can be conditioned already during the provisional period, in order to obtain an ideal aesthetic [19,20,21,22]. If it is certainly true that the height of soft tissues is key [20,22]; to date, however, only few studies have addressed this topic and analyzed the relationship between height and thickness of peri-implant tissues [22,23,24].

Nozawa et al. measured the relationship between height and thickness of the peri-implant tissues, finding a ratio of 1:1.5 [24]; this ratio was exactly the opposite of that measured by Wennström on natural teeth [23]. 

In our present study, the average value of the MH was 3.44 mm (SD 1.28), while the MT mean value was 3.29 (SD 1.46); therefore, the ratio between mucosal height and thickness was found to be 1:1.19.

Nozawa’s measurements were taken on 14 implants after final prosthetic load with an average period of 3 year and 5 months [24]. Compared to Nozawa’s work, in our present study a larger number of implants (33 fixtures) has been investigated. In addition, our measurements were made after one year from provisional prosthetic loading, when the tissue maturation process had already taken place [30,31,32]. 

All these intrinsic differences from Nozawa’s original examination may have affected our results, giving a different ratio in our study; anyway, the relationship between the height and the width is still present in our study, and statistically strong.

However, a further, important difference between our present study and the previous investigation of Nozawa et al. [24] could be represented by the implants used, with particular emphasis for the implant–abutment connection. All the implants analyzed by Nozawa et al. had an internal hexagon with a flat-to-flat connection (Frialit-2^®^ Dentsply, York, PA, USA), while all our samples presented a 5-mm-deep conical connection (10°) combined with an internal hexagon (Anyridge^®^, MegaGen, Gyeongbuk, South Korea), with an integrated platform switching [21,22]. The conical connection and the platform switching of the Anyridge^®^ implants seem to guarantee an excellent quality and quantity of soft tissues, as confirmed by the tissue height and thickness reported here; these tissues appear to be stable along time.

In a recent systematic review, results indicated a reduced occurrence of peri-implantitis and bone loss at the abutment/implant level associated with implants with reduced-diameter platform switching abutment [33]. Extrapolation of data from previous studies indicates that implants with platform switching have shown less inflammation and possible bone loss with the peri-implant soft tissues [33,34,35,36]. Platform switching allows a better protection in terms of vascularization and architecture of the soft tissue around the implant [37]. Moreover, a deep conical connection guarantees higher mechanical stability [38,39] and better seal against microbial penetration, compared to flat-to-flat implant–abutment interfaces [39,40,41,42]. The formation of microgap is minimized in conical connections, providing a reduction of bacterial penetration [38,39]. At this level a better seal is associated with a stabilization of hard and soft peri-implant tissues [38]. 

Hence, the use of an implant system with a conical connection and integrated platform switching, as in the present study, may be advantageous from an aesthetic point of view: since the ratio between height and thickness is more favourable than that of Nozawa et al. [24], there is no need for an excessive tissue thickness to get an adequate height. In other words, since the ratio between height and thickness in our present study is more favourable than that reported by Nozawa et al., [24] adequate tissue height (and therefore better aesthetics) maybe achieved starting from a lower tissue thickness.

This is not a trivial matter, because the final aesthetic outcome of a restoration is strictly related to peri-implant tissue height [39,40,41,42,43]. Clinicians should therefore use an implant system and surgical protocols that allow them to obtain a significant sagittal thickness of the peri-implant tissue. This can be the right way to achieve a vertical height that can be conditioned by the prosthetic restoration, in order to achieve excellent and stable aesthetic outcomes.

The present study has limits, such as the limited number of implants examined, the heterogeneous implant distribution by position and diameter, the fact that only one implant system has been evaluated, the use of an analog method for the evaluation, no radiographic monitor of the implant relations to bone and the limited follow-up time; further studies on a larger sample of patients, using modern digital technologies such as intraoral scanners [44,45] and investigating the variations occurring at tissue level, should be performed, to better elucidate this issue. Moreover the study of the MH/MT ratio could have potentialities referring to the horizontal/vertical dimensions of the biologic width correlated to the implant design. In oder to verify this biological findings a comparative analysis with larger number of implants should be considered for further findings. 

## 5. Conclusions

In the present study, the average value of the MH was 3.44 mm (SD 1.28) while the MT mean value was 3.29 (SD 1.46); therefore, the ratio between mucosal height and thickness was found to be 1:1.19. Our observations, supported by the existing literature, confirmed that there is a constant relationship between width and height of the peri-implant mucosa. However, our results are different from those of Nozawa et al., who found a ratio of 1:1.5 between height and thickness of the peri-implant tissues. The intrinsic differences between our study and that of Nozawa et al. (i.e., the larger sample of implants investigated, the longer follow-up time, and the different implant system used) may have produced such a different ratio. However, one of the reasons for this result could be the different implant system used in our study. In our study, in fact, we have used implants with conical connection and integrated platform switching. Clinicians should use an implant system and surgical protocols that allow them to obtain a significant sagittal thickness of the peri-implant tissue. This can be the proper way to achieve a vertical height that can be conditioned by the prosthetic restoration, in order to achieve excellent and stable aesthetic outcomes.

## Figures and Tables

**Figure 1 dentistry-07-00040-f001:**
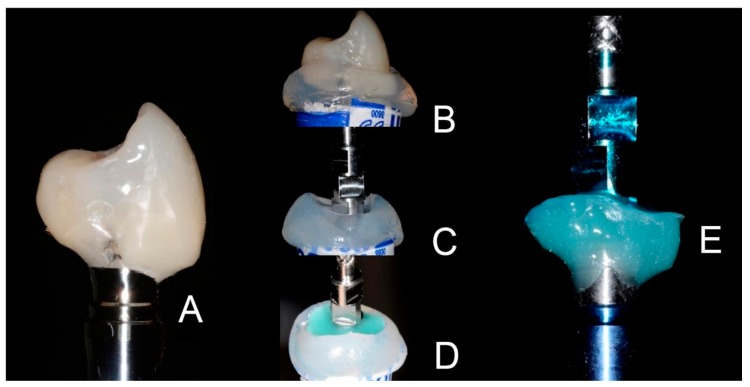
The individual transfer technique consists of replicating the same emergence profile of the provisional crown on the transfer, so as not to induce any further tissue adaptation to the prosthetic margins. The provisional is connected to an analogue (**A**), a silicon impression (Aquasil®, Dentsply Sirona, York, PN, USA) of the assembly is performed (**B**) and, after disconnecting the provisional, a transfer is set in place connected to the analogue sitting in the silicone (**C**). In the gap between the silicon and the transfer some flowable composite is light-cured (**D**). The transfer reproduces the same emergence as the provisional (**E**).

**Figure 2 dentistry-07-00040-f002:**
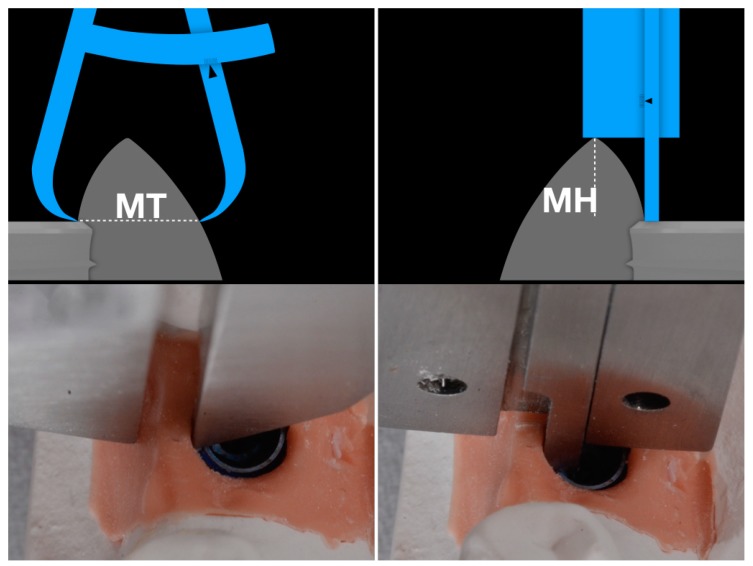
The height of the peri-implant mucosa (MH), calculated from the vestibular shoulder of the implant analogue to the upper gingival margin of the supra-implant tissue according to the implant major axis. The width of tissue (MT) at the connection level is calculated from the vestibular shoulder of the analogue to the external mucosa point perpendicular to the implant major axis. The measurements are taken over the buccal zenith parabola.

**Table 1 dentistry-07-00040-t001:** Individual measurements of mucosal height (MH), mucosal width (MT) and their ratio (MH/MT).

Patient Number	Implant Number	Implant Position	Implant Diameter	Mucosal Height (MH)	Mucosal Width (MT)	Height/Width Ratio (MH/MT)
1	1	36	4 mm	4.5 mm	5.0 mm	0.90
2	2	16	6 mm	4 mm	3.0 mm	1.33
2	3	35	7 mm	5 mm	5.0 mm	1.00
2	4	36	8 mm	5 mm	4.0 mm	1.25
2	5	46	9 mm	4 mm	4.0 mm	1.00
3	6	14	4.5 mm	3 mm	2.0 mm	1.50
4	7	47	4.5 mm	3 mm	3.0 mm	1.00
5	8	26	3.5 mm	3 mm	2.0 mm	1.50
5	9	45	3.5 mm	3 mm	1.0 mm	3.00
5	10	16	4.5 mm	2 mm	1.0 mm	2.00
6	11	36	4 mm	4 mm	5.0 mm	0.80
7	12	47	6 mm	3 mm	4.0 mm	0.75
8	13	13	4 mm	3 mm	1.5 mm	2.00
8	14	11	4.5 mm	3 mm	1.5 mm	2.00
9	15	47	4.5 mm	3 mm	4.0 mm	0.75
10	16	27	6 mm	3 mm	5.0 mm	0.60
10	17	24	4 mm	3 mm	4.0 mm	0.75
10	18	25	4 mm	3 mm	5.0 mm	0.60
11	19	47	4 mm	3.8 mm	3.0 mm	1.26
11	20	35	4 mm	7 mm	6.0 mm	1.16
11	21	37	6 mm	7 mm	6.0 mm	1.16
12	22	45	4 mm	3.5 mm	3.0 mm	1.16
12	23	46	4 mm	3.5 mm	3.5 mm	1.00
13	24	27	4.5 mm	3 mm	2.0 mm	1.50
14	25	35	4 mm	3 mm	1.5 mm	2.00
15	26	25	3.75 mm	3 mm	4.0 mm	0.75
16	27	47	5.5 mm	1 mm	2.0 mm	0.50
17	28	24	4.5 mm	3 mm	2.0 mm	1.50
18	29	37	5.5 mm	1 mm	2.5 mm	0.40
19	30	37	8.5 mm	2.5 mm	3.0 mm	0.83
20	31	14	3.5 mm	4.5 mm	5.0 mm	0.90
20	32	26	5 mm	3 mm	2.0 mm	1.50

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
