# Peer review of "Ratio between Height and Thickness of the Buccal Tissues: A Pilot Study on 32 Single Implants"

_dentistry, 2019, doi:10.3390/dj7020040_

Round 1

Reviewer 1 Report

excellent research work on a very interesting and innovative topic. i suggest publication after a few minor corrections have been performed. i kindly ask the authors to change their paper accordingly.

----title: is ok

----abstract: the following changes are needed:

line 21, "aesthetic outcome is one of the most important.."

line 27, deep in underlined, why? "and integrated platform switching" (not integrate)

line 50, please remove "therefore" 

line 53, outcome (and not outcomes)

----introduction: very well written the following changes are requested:

line 106-110 please give more details about the implants used and particularly the implant-abutment connection featured in the Nozawa et al. study. this information is important for the reader. please expand your introduction in accordance with this suggestion.

line 91-94, please insert the correct reference for Wang et al. (you mention it and no reference for supporting this sentence)

line 103, "soft tissues" and not "soft-tissues"

----methods: are clear and concise, please modify accordingly:

line 120 please insert info about the implant manufacturer

line 125 what do you meen for good oral hygiene?

line 150 prosthetic casts, you mean "stone casts"?

----results: are perfect

----discussion: just a few clarifications 

line 211, platform switching was not "discovered" but "found" from Lazzara et al. please amend

line 220-225, please expand the concept. why the connection of the implants used here should guarantee better results, being beneficial in terms of the final aesthetic outcome? can you reformulate and explain in more detail, by adding another sentence?

----conclusions: fine

----references: complete

----figures: good

Author Response

REVIEWER N°1

excellent research work on a very interesting and innovative topic. i suggest publication after a few minor corrections have been performed. i kindly ask the authors to change their paper accordingly.

RESPONSE- THANK YOU VERY MUCH FOR YOUR COMMENT. WE REALLY BELIEVE THIS WORK CAN REPRESENT A VALUABLE CONTRIBUTION TO THE PRESENT KNOWLEDGE IN THE DENTAL IMPLANT LITERATURE.

----title: is ok

RESPONSE- THANK YOU. WE FOUND A TYPING ERROR IN THE TITLE AND WE HAVE CORRECTED IT, AS SUGGESTED BY REVIEWER N° 2. IN ADDITION, WE HAVE CHECKED AGAIN ALL OUR MODELS AND MEASUREMENTS AND WE DECIDED TO FURTHER EXCLUDE ONE IMPLANT BECAUSE OF A LITTLE IMPREFECTION IN THE STONE CAST. THEREFORE, WE HAVE UPDATED ALL OUR TEXT INCLUDING THE TITLE, CONSIDERING IN OUR ANALYSIS ONLY 32 IMPLANTS (NOT 33). WE HAVE MODIFIED OUR ABSTRACT, OUR RESULTS AND OUR DISCUSSION AND CONCLUSIONS ACCORDINGLY, AS WE HAVE RE-CALCULATED ALL.

----abstract: the following changes are needed:

line 21, "aesthetic outcome is one of the most important.."

RESPONSE- WE HAVE CORRECTED OUR TEXT AS REQUESTED.

line 27, deep in underlined, why? "and integrated platform switching" (not integrate)

RESPONSE- WE HAVE CORRECTED ALL TYPING ERRORS AS REQUESTED.

line 50, please remove "therefore"

RESPONSE- WE HAVE CORRECTED OUR TEXT AS REQUESTED.

line 53, outcome (and not outcomes)

RESPONSE- WE HAVE CORRECTED OUR TEXT AS REQUESTED.

----introduction: very well written the following changes are requested:

line 106-110 please give more details about the implants used and particularly the implant-abutment connection featured in the Nozawa et al. study. this information is important for the reader. please expand your introduction in accordance with this suggestion.

RESPONSE- WE HAVE GIVEN MORE INFORMATION ABOUT THE IMPLANT SYSTEM USED IN THE STUDY OF NOZAWA ET AL. IN THE INTRODUCTION SENTENCE, BY ADDING ONE NEW SENTENCE, SO WE HAVE EXPANDED OUR INTRODUCTION SECTION ACCORDINGLY.

line 91-94, please insert the correct reference for Wang et al. (you mention it and no reference for supporting this sentence)

RESPONSE- WE HAVE INSERTED THE CORRECT REFERENCE.

line 103, "soft tissues" and not "soft-tissues"

RESPONSE- WE HAVE CORRECTED OUR TEXT AS REQUESTED.

----methods: are clear and concise, please modify accordingly:

line 120 please insert info about the implant manufacturer

RESPONSE- WE HAVE INSERTED INFORMATION ABOUT THE IMPLANT MANUFACTURER AS REQUESTED.

line 125 what do you meen for good oral hygiene?

RESPONSE- FOR GOOD LEVELS OF ORAL HYGIENE WE REFER HERE TO A SATISFACTORY PLAQUE CONTROL ACHIEVED THROUGH PROFESSIONAL ORAL HYGIENE SESSIONS (TWICE PER YEAR) PLUS AN ADEQUATE BRUSHING FLOSSING AND DOMESTIC ORAL HYGIENE PROCEDURES. WE HAVE STATED THIS IN THE TEXT.

line 150 prosthetic casts, you mean "stone casts"?

RESPONSE- YES, WE HAVE CORRECTED OUR TEXT ACCORDINGLY

----results: are perfect

RESPONSE- THANK YOU.

----discussion: just a few clarifications

line 211, platform switching was not "discovered" but "found" from Lazzara et al. please amend

RESPONSE- WE HAVE CORRECTED OUR TEXT AS REQUESTED.

line 220-225, please expand the concept. why the connection of the implants used here should guarantee better results, being beneficial in terms of the final aesthetic outcome? can you reformulate and explain in more detail, by adding another sentence?

RESPONSE- WE HAVE EXPANDED THE TEXT HERE REFORMULATION THE SENTENCES IN ORDER TO BETTER CLARIFY THE POTENTIAL ADVANTAGES OF IMPLANTS WITH CONICAL CONNECTION AND PLATFORM SWITCHING..

----conclusions: fine

RESPONSE- THANK YOU.

----references: complete

RESPONSE- THANK YOU.

----figures: good

RESPONSE- THANK YOU.

Reviewer 2 Report

The topic of this manuscript is of interest to the dental community. However, it requires some major corrections to improve the content.

The methodology section is unclear. The patient selection and surgical and prosthetic procedures sections require additional information such as implant placement protocol, position of implants, additional details re. number of males/females, age groups etc.

The figure 1 is very difficult to see with the black background. Was the impression taken using these and if so it requires additional information regarding the procedure including the materials used.

How was the measurement carried out? Do you have the information re. inter- or intra-observer variability?

Results section requires additional information as providing only the overall mean values is insufficient. As presented in Nozawa et al (2006) paper, it would be useful to have individual measurements also included in a table format.

Authors have suddenly introduced the concept of platform switching in the Discussion section. This should be included in the Introduction part.

Conclusions do not match the results seen in the study. As there was no aesthetic evaluation (PES) of the participants, it cannot be concluded that better aesthetic results can be achieved with "adequate" keratinized tissue height associated with less tissue thickness.

Minor corrections.

The title has a spelling error.

There is no reference no.11 in the reference list.

Lines 99 - 117 require some editing to reduce the number of paragraphs with a single sentence only.

Authors need to be consistent with the terminology and replace "fixture" with "implant".

The reference section requires some reformatting in order to be consistent with the journal referencing style.

Author Response

REVIEWER N°2

The topic of this manuscript is of interest to the dental community. However, it requires some major corrections to improve the content.

RESPONSE- THANK YOU VERY MUCH FOR YOUR COMMENT. WE REALLY BELIEVE THIS WORK CAN REPRESENT A VALUABLE CONTRIBUTION TO THE PRESENT KNOWLEDGE IN THE DENTAL IMPLANT LITERATURE. WE HAVE PERFORMED A CAREFUL REVISION OF OUR PAPER AND WE HAVE REPLIED POINT- BY- POINT TO ALL THE CRITICISMS AND QUESTIONS RAISED BY THE REVIEWERS.

The methodology section is unclear. The patient selection and surgical and prosthetic procedures sections require additional information such as implant placement protocol, position of implants, additional details re. number of males/females, age groups etc.

RESPONSE- WE HAVE EXPANDED OUR METHODOLOGY SECTION BY PROVIDING MORE DETAILS ON THE METHOD USED FOR MEASUREMENTS. IN ADDITION WE HAVE EXPANDED OUR RESULT SECTION GIVING MORE ABOUT THE PATIENTS SELECTED, AND THE IMPLANTS PLACED. WE HAVE CHECKED AGAIN ALL OUR MODELS AND MEASUREMENTS AND WE DECIDED TO FURTHER EXCLUDE ONE IMPLANT BECAUSE OF A LITTLE IMPREFECTION IN THE STONE CAST. THEREFORE, WE HAVE UPDATED ALL OUR TEXT INCLUDING THE TITLE, CONSIDERING IN OUR ANALYSIS ONLY 32 IMPLANTS (NOT 33). WE HAVE MODIFIED OUR ABSTRACT, OUR RESULTS AND OUR DISCUSSION AND CONCLUSIONS ACCORDINGLY, AS WE HAVE RE-CALCULATED ALL.

The figure 1 is very difficult to see with the black background. Was the impression taken using these and if so it requires additional information regarding the procedure including the materials used.

RESPONSE- THE FIGURE IS NOW IN COLOUR, IN ORDER THE READER CAN BETTER UNDERSTAND THE PROCEDURES, AS REQUESTED, AND A DETAILED DESCRIPTION OF THE IMPRESSION TECHNIQUE IS GIVEN IN THE LEGEND OF FIGURE 1, IN THE TEXT AS WELL. THE COMMERCIALLY USED MATERIAL WAS AQUASIL FROM DENTSPLY SIRONA.

How was the measurement carried out? Do you have the information re. inter- or intra-observer variability?

RESPONSE- WE HAVE PROVIDED MORE DATA ABOUT THE MEASUREMENT METHOD.

Results section requires additional information as providing only the overall mean values is insufficient. As presented in Nozawa et al (2006) paper, it would be useful to have individual measurements also included in a table format.

RESPONSE- WE HAVE EXPANDED THE RESULT SECTION BY ADDING THE TABLE WITH ALL INDIVIDUAL MEASUREMENT. WE HAVE NAMED IT AS TABLE 1.

Authors have suddenly introduced the concept of platform switching in the Discussion section. This should be included in the Introduction part.

RESPONSE- WE HAVE INTRODUCED THE CONCEPT OF PLATFORM SWITCHING BY LAZZARA ET AL. IN THE INTRODUCTION; THE CONCEPT IS THEN DEVELOPED IN THE DISCUSSION SECTION. AS A CONSEQUENCE, WE HAD TO RE-NUMBER ALL OUR REFERENCES.

Conclusions do not match the results seen in the study. As there was no aesthetic evaluation (PES) of the participants, it cannot be concluded that better aesthetic results can be achieved with "adequate" keratinized tissue height associated with less tissue thickness.

RESPONSE- WE HAVE REDUCED OUR CONCLUSIONS SECTION, AS SUGGESTED, WE HAVE MOREOVER REMOVED THAT MISLEADING SENTENCE.

Minor corrections.

The title has a spelling error.

RESPONSE- THANK YOU. WE FOUND ONE TYPING ERROR IN THE TITLE AND WE HAVE CORRECTED IT, AS SUGGESTED. IN ADDITION WE HAVE EXCLUDED ONE MORE IMPLANT, BECAUSE OF A LITTLE IMPERFECTION IN THE STONE CAST. WE HAVE CHECKED AGAIN ALL OUR MODELS AND MEASUREMENTS AND WE DECIDED TO FURTHER EXCLUDE ONE IMPLANT BECAUSE OF A LITTLE IMPREFECTION IN THE STONE CAST. THEREFORE, WE HAVE UPDATED ALL OUR TEXT INCLUDING THE TITLE, CONSIDERING IN OUR ANALYSIS ONLY 32 IMPLANTS (NOT 33). WE HAVE MODIFIED OUR ABSTRACT, OUR RESULTS AND OUR DISCUSSION AND CONCLUSIONS ACCORDINGLY, AS WE HAVE RE-CALCULATED ALL.

There is no reference no.11 in the reference list.

RESPONSE- THANK YOU. WE HAVE ADDED REFERENCE NUMBER 11 AS REQUESTED. 

Lines 99 - 117 require some editing to reduce the number of paragraphs with a single sentence only.

RESPONSE- WE HAVE REDUCED THE LENGTH OF THIS PARAGRAPH AS REQUESTED.

Authors need to be consistent with the terminology and replace "fixture" with "implant".

RESPONSE- WE DO NOT USE THE TERM “FIXTURE” ANYMORE. WE HAVE REMOVED THAT TERM FROM THE ABSTRACT, WHERE IT WAS USED TWICE; WE HAVE REMOVED IT IN THE TEXT. 

The reference section requires some reformatting in order to be consistent with the journal referencing style.

RESPONSE- THE ONLY THING THAT WAS NOT IN FULL ARCCORDANCE WITH THE INSTRUCTIONS FOR THE AUTHORS WAS THE PRESENCE OF THE ISSUE OF THE JOURNAL, IN PARENTHESIS (). WE HAVE REMOVED IT. WE HAVE CORRECTED THE STYLE OF OUR REFERENCES USING THE RECENTLY PUBLISHED PAPER AS EXAMPLE OF PERFECT FORMAT:

Percival T, Edwards J, Barclay S, Sa B, Majumder MAA.

Early Childhood Caries in 3 to 5 Year Old Children in Trinidad and Tobago.

Dent J (Basel). 2019 Feb 7;7(1).

THIS PAPER IS PUBLISHED ALREADY AND THE REFERENCES STYLE IS IDENTICAL TO OUR STYLE.

Round 2

Reviewer 1 Report

the authors have carefully revised their manuscript that is now acceptable for publication.

Author Response

THANK YOU

Reviewer 2 Report

Happy with the amended version.

Author Response

THANK YOU